# An Individually Tailored Program to Increase Healthy Lifestyle Behaviors among the Elderly

**DOI:** 10.3390/ijerph191711085

**Published:** 2022-09-04

**Authors:** Sharon Barak, Tzlil Rabinovitz, Achinoam Ben Akiva-Maliniak, Rony Schenker, Lian Meiry, Riki Tesler

**Affiliations:** 1Department of Nursing, School of Health Sciences, Ariel University, Ariel 4076414, Israel; 2Department of Pediatric Rehabilitation, The Edmond and Lily Safra Children’s Hospital, The Chaim Sheba Medical Center, Ramat-Gan 5290002, Israel; 3JDC-Eshel, American Jewish Joint Distribution Committee, Jerusalem 9103401, Israel; 4Program in Gerontology, Faculty of Health Sciences, Ben-Gurion University, Beer Sheeva 8410501, Israel; 5Department of Health Systems Management, School of Health Sciences, Ariel University, Ariel 4076414, Israel

**Keywords:** elderly, healthy lifestyle, healthy behaviors, health promotion programs, compliance

## Abstract

A healthy lifestyle among the elderly is associated with improved health. However, many older adults are not engaging in such behavior. The purpose of the study was to develop an individually tailored online/telephone program to increase healthy lifestyle behaviors among community-dwelling elderly people. The program includes individually tailored healthy lifestyle recommendations based on participants’ functional level. Community-dwelling elderly people aged 60+ years (*n* = 77; mean age: 72.98 ± 6.49) participated in the study. Significant associations were observed between health promotion activities and health status (r = 0.23, *p* = 0.04) and physical functional level (r = 0.44, *p* < 0.001). Twenty-seven percent of participants claimed that they learned “a lot” of new things about themselves, and 31% claimed that the recommendations received were new to them. Most participants engaged in the recommendations at least 1–2 times a week. Regression analyses showed that barriers significantly predicted reduced compliance with the health-related recommendations received (adjusted R^2^ = 0.18). The main barrier for compliance was inaccessible information about services (32.46% of the participants). The most prevalent facilitator for compliance with the recommendation was health behavior motivation (59.74% of the participants). In conclusion, this study provided evidence on the effectiveness of a multicomponent tailored intervention program among the elderly in increasing health-related knowledge and behavior about the recommendations.

## 1. Introduction

Today, most people can expect to live into their 60s and beyond [1]. This demographic shift has profound implications at both the individual and the societal levels. At the individual level, aging is associated with a gradual decrease in health and function [2]. At the societal level, many of the oldest individuals require some form of long-term care [3]. However, health and functional status are influenced not only by chronological age, but also by healthy lifestyle behaviors [4,5,6]. Therefore, engaging in healthy lifestyle behaviors can prevent or minimize poor physical and mental health in old age [7,8,9].

Despite the beneficial effects of healthy lifestyle behaviors, the majority of older adults do not adhere to the various recommendations for such behaviors [10,11,12,13,14]. Therefore, safe and effective ways to promote healthy lifestyle behaviors as people age are necessary. Accordingly, numerous studies on programs aimed at promoting healthy lifestyle behaviors have been conducted (e.g., programs to increase physical activity) [15].

Despite rich research in healthy lifestyle promotion, published interventions that target older people usually focus on a specific healthy lifestyle behavior (e.g., physical activity), and studies aiming at improving various aspects of healthy lifestyle behaviors are not as common. Moreover, in order to promote healthy lifestyle behavior, it is helpful to develop programs driven by theoretical frameworks, such as the ecological model. The ecological model is multidimensional, and it focuses on three general levels: individual, interpersonal, and community factors [16,17]. Reference [18] categorized the ecological system into a microsystem, mesosystem, and macrosystem. The microsystem includes an individual’s social, biological, and psychological settings; the mesosystem includes small groups (e.g., family) that surround the individual; and the macrosystem includes the community and country. When developing health promotion programs, it is important to consider the different systems encompassing and surrounding the elderly, as well as their individual characteristics, for example, learning about both the individual and environmental facilitators and barriers to adopting new healthy lifestyle behaviors.

Accordingly, the main purpose of the study was to develop an individually tailored program to increase healthy lifestyle behaviors among the community-dwelling elderly. The program is unique as it includes multiple aspects of healthy lifestyle behaviors (e.g., routine health examinations, physical activity, nutrition, cognition, socioemotional health, and more). Moreover, the healthy lifestyle recommendations given to participants are individually tailored to the participants’ physical, cognitive, and psychological condition. Moreover, to achieve a widespread impact of the program, participants received recommendations about services available in their local community. Information regarding local services might increase one’s ability to implement the activities recommended to them and reduce environmental system barriers. More specifically, the study examined: (1) participants’ perceptions on and compliance with the healthy lifestyle-related services recommended, (2) facilitators and barriers to implementing the health-related recommendations received, and (3) factors related to and predicting usage of the recommended health-related services.

## 2. Materials and Methods

### 2.1. Research Participants

#### 2.1.1. Inclusion Criteria

(1) Sex: men and women, (2) Age: 60+ years old, (3) Living arrangement: living in the community, (4) Digital literacy level: with or without digital literacy.

#### 2.1.2. Exclusion Criteria

Excluded from this study were individuals living in nursing home facilities and/or with severe functional decline (i.e., fully dependent on another person for conducting basic activities and instrumental activities of daily living).

Study participants were recruited via the following channels: (1) through coordinators of the “support-community” program (a program designed to give support for the needs of the elderly living independently in the community); (2) through managers of social clubs for the elderly in local authorities; and (3) via social networks (e.g., groups of retirees in local authorities). All participants gave their consent to partake in the study and signed the informed consent prior to program’s initiation.

A total of *n* = 124 attended the first program’s meeting. Out of the *n* = 124 participants, *n* = 77 continued and completed the program (62.09% of participants). No statistical differences between those who completed (*n* = 77) and did not complete the study (*n* = 47) were observed in demographic characteristics (i.e., age, sex, family status, education level, and income).

### 2.2. Research Procedure

#### 2.2.1. About the Program

The program, “Active-Hug”, is a cooperation between JDC-ESHEL and The Ruth Vrobel Foundation. JDC-ESHEL is Israel’s social research and development incubator, tasked with developing comprehensive responses to the complex challenges faced by Israeli society with the aging of its population. The organization brings together and serves as an “honest-broker” and conveyor among government ministries, prominent stakeholders, and professionals involved in the care of older adults. Many of the 1.2 million individuals aged 65+ living in Israel benefit directly and indirectly from a rich variety of services and programs designed and developed by JDC-ESHEL. JDC ESHEL functions under the auspices of the Joint Distribution Committee (JDC), an American-based NGO, which provides relief, rescue, and reconstruction services to Jewish and nonJewish clients around the world. The Ruth Vrobel Foundation established in 2003 works to improve the quality of life of people with chronic diseases and their families. The foundation creates an additional and complementary layer for public health and community care services in Israel through identifying patient needs and producing a series of comprehensive solutions nationwide through partnerships with major stakeholders in the community.

The program’s goal is to increase among older adults a healthy lifestyle and the consumption of health promotion-related services and functional independence, by providing recommendations tailored to the individual level of functioning. The program operated during the coronavirus pandemic 2019 (August 2020 to March 2021), when the older population was confined to their homes. Therefore, during this period, the meetings between the student and the participant were conducted remotely via a computer. Each student accompanied two or three older adults during the program.

#### 2.2.2. Program Components

The program included two main components: (1) a digital system for functional assessment and (2) providing recommendations for health promotion recommendations in accordance with the functional assessment results.

Digital System for Functional Assessment.

The functional assessment includes four different areas: (a) health–medical condition, (b) physical function, (c) cognitive function, and (d) emotional–social function (Figure 1).

Part A—Health–medical status: This section includes nine questions related to the participants’ health condition and morbidity. The consisted of questions that may indicate a state of frailty (e.g., unintentional weight loss of more than 5 kg [19] and common morbidity conditions in old age (e.g., heart disease and stroke). Based on the total score, participants were categorized as having low, medium, or high health–medical condition status.Part B—Physical function: This part of the questionnaire includes 11 questions. Some of the questions deal with the person’s general level of functioning (for example, walking ability, ability to perform daily actions, instrumental actions of daily actions), and history of falls (for example, number of falls in the last year). Based on the total score, participants were categorized as having low, medium, or high physical function status and/or at risk for falling.Part C—Cognitive function: Cognitive function was assessed using the Alzheimer’s disease-8 (AD8) questionnaire [20]. In community-residing older adults, AD8 shows excellent internal reliability (Cronbach alpha = 0.84), excellent internal consistency, and strong interrater reliability and stability [21]. A score of 2 or more is commonly used as the cut-off score to indicate dementia [22].Part D—Emotional–social function: Emotional–social function was assessed using the Patient Health Questionnaire-2 (PHQ-2). This questionnaire includes two questions that deal with the frequency of onset of symptoms of depression and discomfort on a scale ranging from zero (not at all) to three (almost every day). The sensitivity and specificity of the assessment tool in identifying clinical depression is 74 and 91% with a score of three or higher [23]. Accordingly, from this part of the questionnaire, the person’s emotional state was classified into one of two levels—there may be emotional problems or no emotional problems.

Moreover, as part of the assessment questionnaire, participants were asked about their desires and needs in various aspects (e.g., transportation assistance). These questions were used to refine the recommendations of services, according to the older person’s wishes and needs. The digital system of functional assessment can be either filled out independently or mediated by an intermediary and an accompanying person (i.e., the nursing student).

The aforementioned assessment scale was developed according to the modified Delphi model [24]. The various development stages of the tool included: (1) literature review, (2) recruitment of a team of experts, (3) meetings with the team of experts, (4) presenting the assessment tool to other professionals outside the team of experts, (5) testing the reliability and validity of the tool—pilot study, and (6) making corrections based on the reliability and validity examinations.

Providing recommendations for health promotion services according to the functional evaluation results.

Health promotion recommendations were given to the participant based on his/her functional level and his/her preferences. For example, in the health–medical condition assessment, participants who reported an unintentional weight loss were referred for further evaluation to their general practitioner and to a nutritionist. In the physical function domain, participants with a high risk of falling were referred, among others, to supervised physical activity programs. In the cognitive domain, participants with cognitive difficulties were recommend, among others, to cognitive-enhancing community-related activities. Finally, in the emotional–social function assessment, participants with problems in this domain were referred to nearby group activities for the elderly.

#### 2.2.3. The Training Process of the Program’s Staff

All the students who participated in the “Active-Hug” program were nursing students. The online training and instruction set (total 12 h) included four sessions, two hours each at the beginning of the program and three instructional sessions later in the program. Training topics included knowledge of the old age—epidemiology, concepts, changes in functional level, and healthy lifestyle. Moreover, students received tools to help them with the conversation with the elderly (e.g., call scripts).

#### 2.2.4. Description of the Meetings between the Students and the Elderly

Each student accompanied via Zoom/phone 1–3 elderly people during the program and had six sessions/conversations with each one of them.

First conversation—Acquaintance with the program (e.g., background of the program and its goals).Second conversation—Presentation of the functional assessment questionnaire and determination of how to complete the questionnaire (independently or with the help of the student). Those who were capable to complete the questionnaire independently were instructed to do so until the following conversation.Third conversation—For those who completed the assessment independently, the student and the elderly person discussed the participant’s experience with the questionnaire. For those who needed help with the questionnaire, the students filled it in with the elderly person.Fourth conversation—A joint review of the assessment’s recommendations, selection of recommendations for implementation, and prioritizing them by the elderly person. Finally, construction of a plan for the implementation of the recommendations.Fifth conversation—Identifying factors that promote and delay the implementation of the recommendations. In this phase, the students also encouraged and motivated the participant to comply with the recommendations given to and selected by them.Sixth conversation—Monitoring compliance with the implementation of the selected recommendations.

### 2.3. Outcome Measures

#### 2.3.1. Demographic and Functional Characteristics

The following demographic characteristics were assessed: age, sex, family status, educational level, and income. Functional characteristics were obtained from the functional assessment tool described in the procedures section.

#### 2.3.2. Participants’ Perception on the Health-Related Recommendations They Received

Upon completion of the functional assessment, study participants received healthy lifestyle recommendations. Participants’ opinion regarding how much they learned from the recommendations and the motivational effects of the recommendations were assessed using four questions on a three Likert scale: ‘not at all’, ‘somewhat’, or ‘a lot’.

#### 2.3.3. Healthy Lifestyle Recommendations and Implementation

From the list of heathy lifestyle recommendations provided to the elderly, the study participants chose a total of three recommendations for implementation. Recommendations were from one or more of the four functional domains assessed. During the student–elderly person conversations, the implementation of the recommendations chosen was established using a scale ranging from ‘never conducted’ to ‘conducted every day’.

#### 2.3.4. Facilitators and Barriers to Healthy Lifestyle Recommendations Implementation

Study participants were given a list of seven facilitators and seven barriers (four personal facilitators and three environmental facilitators). Participants could select more than one facilitator/barrier. The list of facilitators and barriers was constructed based on a literature review that the research team conducted on the topic.

### 2.4. Data Analyses

#### 2.4.1. Demographic and Functional Characteristics of Study Participants and Attrition Group

Descriptive statistics were used to describe the demographic and functional characteristics of the study group. Moreover, demographic characteristics of the study group and attrition group were compared using independent *t*-tests and chi-squared tests.

#### 2.4.2. Participants’ Perception on and Compliance with the Healthy Lifestyle-Related Services Recommended

First, the prevalence of the different types of health-related services that study participants were interested in receiving information about was calculated. In the next step, participants’ perception on the health-related recommendations they received was calculated and compared using chi-squared test. More specifically, the percentage of participants reporting that they did not learn ‘at all’ or ‘somewhat’ from the recommendations was compared to the percentage of those reporting that they ‘learned a lot’ from the recommendations. Finally, the prevalence of compliance in the three health-related recommendations received was calculated and compared using chi-squared test.

#### 2.4.3. Facilitators, Barriers, and Predictors of Compliance with Health-Related Services

The prevalence of each of the seven facilitators and barriers to compliance with the health-related recommendations received was calculated. In addition, factors (demographic variables, functional profiles, and facilitators and barriers) related to the frequency of compliance with the first, second, and third health-related recommendations received were examined with Pearson’s correlations. Variables that statistically significantly correlated with the level of compliance were entered into a multiple regression analysis model in order of the correlation’s strength. All independent variables were checked for multicollinearity using variance of inflation factor >10 [25]. The criterion for inclusion in the model was an α level of 0.05, and the exclusion criterion was an α level of 0.10. Post hoc power analysis for the regression analysis was conducted. The regression analysis consisted of only three variables. Based on the mean correlations of the predictors with compliance, partial R^2^ was 0.33 (i.e., medium effect size). Based on these statistical values and a sample size of 77, the power to predict compliance with health-related recommendations was 0.95.

All data analyses, except for power analysis, was conducted using SPSS version 25 (SPSS Inc., Chicago, IL, USA). In all analyses, the significance level was set at 0.05 (two-tailed). Power analysis was conducted using G*Power (version: 3.0.10; Aichach, Germany) [26].

## 3. Results

### 3.1. Demographic and Functional Characteristics of Study Participants

The participants’ mean age was 72.98 ± 6.49. In terms of functional characteristics, most study participants had moderate-to-high general health and physical function with no cognitive or emotional difficulties (Table 1).

### 3.2. Involvement in Health Promotion Activities

Most participants regularly engage in health promotion activities (*n* = 59, 76.61% of the sample). An additional 11.68% of the sample (*n* = 9) is not involved in any health promotion activities; however, they are interested in initiating such activity. Finally, only 11.68% of the sample (*n* = 9) is not interested at all in engaging in health promotion activities. In comparison to females, a higher prevalence of engagement in healthy behavior activities was observed among males (95.5% of males vs. 68.6% of females). Significant associations were observed between the level of engagement in health promotion activities and health status (r = 0.23, *p* = 0.04) and physical functional level (r = 0.44, *p* < 0.001). No statistically significant associations were observed with the other functional status variables, namely, falls, cognitive, and emotional (r range: 0.11–0.17; *p* > 0.05).

### 3.3. Types of Services Participants Are Interested in Receiving Information About

Four health-related services were evaluated, namely, general, physical, cognition, and emotional health. The interest in receiving information about the four health domains ranged from 38% (cognition) to 51% (general health; chi-squared = 2.60, *p* = 0.10; Figure 2).

### 3.4. Participants’ Perception on the Health-Related Recommendations They Received

Overall, 27% of study participants claimed that they learned ‘a lot’ of new things about themselves. Moreover, 71% of participants reported that the recommendations they received increased their motivation to engage in healthy behaviors (Table 2).

### 3.5. Frequency of Compliance with the Recommended Health-Related Recommendations

The prevalence of compliance with the recommendations ‘3–6/week’ was statistically significantly greater than of ‘never’ and ‘1–2/week’ (*p* < 0.05) and ranged from 37% (with the first recommendation) to 38% (with the second and third recommendations; Table 3).

In the next step, for the first recommendation, participants were grouped into those with a low level of compliance with the recommendations (responses “never” and “1–2/week”; *n* = 28) and to those with a high compliance level (responses “3–6/week” and “every day”; *n* = 49). Independent t-tests showed a statistically significant difference between group differences in two outcomes: facilitators and barriers scores. More specifically, the high compliance group, in comparison to the low compliance group, presented more facilitators (4.52 + 1.21 and 2.34 + 1.11, respectively, t = −8.09, *p* < 0.0001) and fewer barriers (3.11 + 1.11 and 5.31 + 1.10, respectively, t = 8.41, *p* < 0.0001).

### 3.6. Facilitators and Barriers to Compliance with the Health-Related Recommendations Received

The most prevalent facilitators with >30% of the sample reporting it, were mainly personal, namely, ‘healthy behavior motivation’, ‘sense of self-efficacy’, and ‘being heathy’. One environmental factor was also cardinal, namely, ‘help and encouragement of family members and friends. One barrier was reported by a considerable percentage of the sample (>30% of the sample), namely, the environmental barrier ‘inaccessible information about services’ (32% of the sample). For additional information (Table 4).

### 3.7. Factors Associated with and Predicting Compliance with the Health-Related Recommendations Received

Statistically significant associations were observed only with total facilitators (r range: 0.21–0.24; *p* < 0.05) and barriers scales (r range: −0.26 to −0.29, *p* < 0.05). Therefore, the regression analyses for predicting compliance with the health-related recommendations received consisted only of three variables: sex, facilitators, and barriers total score. Only the barriers total score was found to be a statistically significant predictor (betta coefficient= −1.20; standard error = 0.20; t-value = −3.20; *p* = 0.01). Overall, the model explained 18% of the observed variability in compliance with health-related recommendations (adjusted R^2^ = 0.18; F-ratio = 2.54; *p* = 0.01).

## 4. Discussion

This study implemented a program to increase healthy lifestyle behaviors among community-dwelling elderly people. The significance of this program is that it is individually tailored to each participant based on his/her functional level, personal desires, and availability of services in his/her community. Moreover, the program did not focus on one particular healthy lifestyle behavior but rather consisted of multiple aspects of healthy lifestyle behaviors. Overall, the program was effective, as it increased participants’ knowledge on recommended healthy lifestyle behaviors, increased the motivation to engage in healthy lifestyle behaviors, and increased healthy lifestyle behaviors conducted by the participants.

The program was conducted by nursing students. Including nursing students in health promotion programs for the elderly is of special importance as nurses in the municipality are one of the foundations of a country’s primary health care [27]. This means that a nurse’s role is shifting from that of traditionally following physicians’ orders to a more expanded role with responsibility for preventive care [28]. Therefore, considering that the current study showed that more than 50% of the participants were interested in receiving health promotion recommendations on more than one health-related domain, it is important to provide nursing students with a broad knowledge needed for health promotion among the elderly [29]. For example, in a systematic review, insufficient knowledge, among others, was a barrier to the uptake of new nursing roles in practice [30].

The current study also aimed at exploring facilitators and barriers to compliance with the health-related recommendations received. Prevalent facilitators were health and help and encouragement of family/friends. Similarly, in a recent study by Huang et al. (2022) [31], facilitators associated with physical activity participation among retired adults were physical and mental health and socioemotional factors. In the current study, self-efficacy was also a strong facilitator. Those with higher levels of self-efficacy were more likely to try a particular task compared to those with lower levels of self-efficacy [32]. One of the ways to increase self-efficacy is via increasing knowledge [31]. Therefore, health promotion programs should make the effort to increase knowledge on health promotion. However, efforts should also be made to guide older adults to convert the knowledge into concrete action plans [33].

Regarding barriers to compliance with health-related recommendations, one barrier was especially prominent, namely, inaccessible information about services. This barrier was not surprising as often information regarding services is mainly available through the Internet. Internet use has become commonplace, but not necessarily among seniors. Moreover, older adults that use computers and the Internet often struggle with a greater number of user errors and require more time to accomplish their goals on the computer than their younger counterparts [34]. Therefore, there is need to provide information regarding health promotion and health promotion services also in non-internet-related manners and to provide services aiming at improving the ability of elderly people to use the computer and the Internet [35].

This study has implications for public health policy. More specifically, from the study, it appears that the elderly are interested in receiving information concerning geographically available health services. Moreover, receiving personally tailored recommendations increases one’s knowledge and motivation to engage in healthy behaviors. Therefore, policy makers should strive to: (1) make health promotion recommendations readily available to the elderly and (2) map health-related services in specific geographical locations. Reading the first recommendation, in comparison to younger individuals, older individuals are more likely to have chronic conditions; their functional capacity is often limited, and they are more likely to experience social isolation. This implies that knowledge regarding health promotion behaviors has to account for the aforementioned factors and requires the involvement of individualized approaches. For example, in European countries, health promotion programs for older people are mainly implemented by primary health care providers and nurses. NGOs, self-governing public authorities, and voluntary organizations are also involved in such health promotion programs. However, as such programs sometimes lack sustainable sources of funding [36], at the health policy level there might be a need for governmental incentives for organizations that implement such health promotion programs. Regarding the second point of mapping health-related services, current methods for mapping health-related services are resource-intensive and often fail to triangulate government data with nongovernmental data and services [37]. Therefore, it might be recommended for policy makers to develop a specific approach to map health promotion services for the elderly. For example, within different population, such as adults with attention deficit and hyperactivity disorder, a mapping method was developed that consisted of the following steps: (1) defining the target service, (2) identifying key informants, (3) designing a survey, (4) data collection, (5) data analysis, (6) communicating findings, and (7) hosting and updating the service map. In this mapping program, patients and members of the public were involved in the various mapping phases [37]. Similar mapping programs may be relevant for mapping health promotion services to the elderly.

This study has some strengths. First, the critical contribution of our study is providing a valuable intervention program utilizing an individually tailored approach to enhance health promotion behaviors within the community. Second, the program utilized a special process to map services available in the community. Third, the intervention was multidisciplinary meeting a vast array of the elderly’s needs. Nonetheless, several limitations should be considered when interpreting our results. First, the study did not use a randomized controlled trial design. Second, our study did not establish long-term effects. Third, this study was conducted during the coronavirus pandemic 2019 when participation in various activities was restricted. Comparison data on health behavior habits pre the pandemic in the country are not available. In addition, excluded from this study were individuals with severe functional decline. Hence, we cannot generalize our findings to other populations.

## 5. Conclusions

In conclusion, this study provided evidence on the effectiveness of a multicomponent (i.e., health, physical, cognitive, and emotional–social) individually tailored intervention program among community-dwelling elderly people in increasing knowledge on health-related recommendations. In addition, most participants engaged in the recommendations provided to them at least 1–2/week. The main facilitators for compliance were healthy behavior motivation, sense of self-efficacy, being heathy, and help and encouragement of family members and friends. The main barrier for compliance was inaccessible information about services. However, only barriers predicted compliance.

## Figures and Tables

**Figure 1 ijerph-19-11085-f001:**
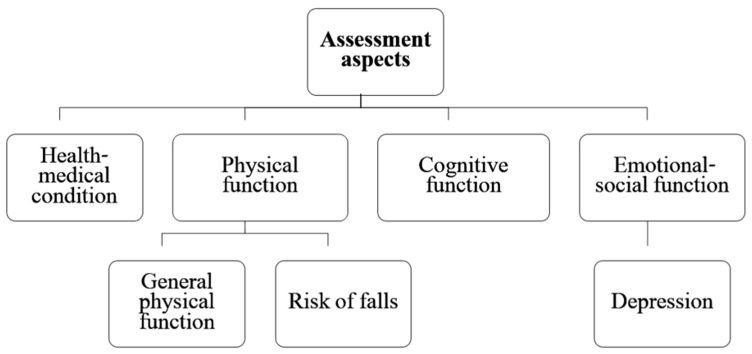
Functional assessment domains and subdomains.

**Figure 2 ijerph-19-11085-f002:**
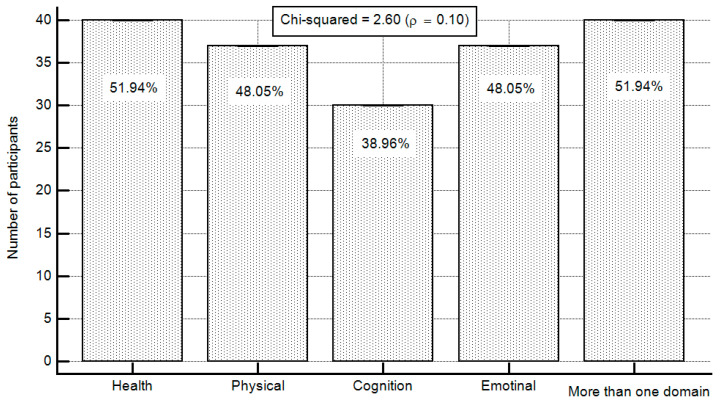
Types of health-related services study participants are interested in receiving information about.

**Table 1 ijerph-19-11085-t001:** Demographic and functional characteristics of study participants (*n* = 77).

			Mean (SD) (Range) or *n* (%)
Demographic characteristics	Age, years: mean (SD)		72.98 (6.49) (60–87)
	Sex: *n* (%)	Females	53 (68.8)
		Males	24 (31.2)
	Family status: *n* (%)	Married	32 (41.6)
		Single	6 (7.8)
		Divorced	8 (10.4)
		Widow	31 (40.3)
	Education: *n* (%)	High school	28 (36.4)
		Tertiary education	12 (15.6)
		Academic	37 (48.1)
	Income: *n* (%)	Below average	39 (50.6)
		Average	28 (36.4)
		Above average	10 (13.0)
Functional characteristics	General health: *n* (%)	Low	12 (15.6)
		Moderate	32 (41.6)
		High	33 (42.9)
	Physical: *n* (%)	Low	14 (18.2)
		Moderate	25 (32.5)
		High	38 (49.4)
	Falls risk: *n* (%)	At risk	52 (67.5)
		Not at risk	25 (32.5)
	Cognitive: *n* (%)	Difficulties	16 (20.8)
		No difficulties	61 (79.2)
	Emotional: *n* (%)	Difficulties	9 (11.7)
		No difficulties	68 (88.3)

Notes: SD, standard deviation; data on functional characteristics were established at the second meeting and therefore were not available for the attrition group.

**Table 2 ijerph-19-11085-t002:** Participants’ perception on the health-related recommendations they received (*n* = 77).

Questions		Not at All/Somewhat: *n* (%)	A Lot: *n* (%)	Chi-Squared (*p* Value)
Learning from the recommendations	Did you learn new things about yourself from the results?	56 (72.72)	21 (27.27)	30.98 (<0.001)
	To what extent were the recommendations you received new to you?	53 (68.83)	24 (31.16)	20.94 (<0.001)
Recommendations and motivation	To what extent did the recommendations you receive increase your motivation to engage in healthy behaviors?	22 (28.57)	55 (71.42)	28.29 (<0.001)
	To what extent did the recommendations you receive decrease your motivation to engage in healthy behaviors?	68 (88.31)	9 (11.68)	90.72 (<0.001)

**Table 3 ijerph-19-11085-t003:** Frequency of compliance with the recommended health-related recommendations (*n* = 77).

Frequency of Compliance with the Recommendations	Never: *n* (%)		1–2/Week: *n* (%)	3–6/Week: *n* (%)	Every Day: *n* (%)	Chi-Squared (*p* Value)
Recommendation number	1	11 (14.28) ^c^	17 (22.07) ^c^	29 (37.66) ^a,b^	20 (25.97)	10.87 (0.001)
	2	13 (16.88) ^c^	13 (16.88) ^c^	30 (38.96) ^a,b^	21 (27.27)	9.26 (0.002)
	3	14 (18.18) ^c^	13 (16.88) ^c^	30 (38.95) ^a,b^	20 (25.97)	9.25 (0.002)

Notes: ^a^, statistically significantly different from “Never” (*p* < 0.05; 2-tailed); ^b^, statistically significantly different from “1–2/week” (*p* < 0.05; 2-tailed); ^c^, statistically significantly different from “3–6/week” (*p* < 0.05; 2-tailed).

**Table 4 ijerph-19-11085-t004:** Facilitators and barriers to compliance with the health-related recommendations received (*n* = 77).

Factor		Item Description	*n* (%)
Facilitators	Personal	Healthy behavior motivation	46 (59.74)
		Sense of self-efficacy	40 (51.94)
		Defining a working plan	22 (28.57)
		Being healthy	34 (44.15)
	Environmental	Help and encouragement of family members and friends	35 (45.45)
		Respecting attitude from health care providers and office holders	12 (15.58)
		Accessible information about services	2 (2.59)
Barriers	Personal	Lack of motivation	4 (5.19)
		Lack sense of self-efficacy, independence, and hope	8 (10.38)
		Unfocused working plan	12 (15.58)
		Being unhealthy	12 (15.58)
	Environmental	Lack of help and encouragement of family members and friends	2 (2.59)
		Unrespecting attitude from health care providers and office holders	8 (10.38)
		Inaccessible information about services	25 (32.46)

Notes: Both “Facilitators” and “Barriers” scales’ internal reliability was acceptable (Cronbach alpha = 0.85 and 0.88, respectively); values represent average number of participants in recommendations 1, 2, and 3.

## Data Availability

The data presented in this study are available on request from the corresponding author. The data are not publicly available due to ethical restrictions.

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
