# Peer review of "An Individually Tailored Program to Increase Healthy Lifestyle Behaviors among the Elderly"

_ijerph, 2022, doi:10.3390/ijerph191711085_

Round 1

Reviewer 1 Report

I would like to thank the authors for this research. This is a very interesting article. The aims of the study was to develop an individually tailored program to increase healthy lifestyle behaviors among dwelling community elderly. 

The research is well designed. However, some modifications are needed in order to improve the quality of the paper. 

One point in regard to the results:

Table 1. Why are you analyzing your research two groups (study group vs. attrition group)? I propose to include only information about the group loss and eventual causes of the study group in the methodological part. 

Given this journal addresses public health, the authors need to discuss the public health implications and/or provide recommendations for strategies or policies that would be informed by these outcomes. 

Also point regarding the funding (line 379), to remove: Please add.

Author Response

Thank you very much for your time reviewing the manuscript and for the informative comments. Attached is a document with description of the changes we have made to the manuscript.

Reviewer 2 Report

The paper is well-written and schematized. The article follows a logical structure, first giving an overview of the literature, explaining the methodology, presenting the results, and discussing them. I liked the research procedure part because it is very well explained and detailed. However, I would have appreciated it if there was a little introduction about the Joint-Eshel and The Ruth Vrobel Foundation. Also, I would like to ask the authors if they have tried to make a comparison with other years. The habits of the elderly may be a bit static due to the time of coronavirus

Regarding the methodology, I would have divided the sample into those who followed the recommendations and those who did not and compared it with other methods, eg MGCFA which analyzes the similarities between the two models. In addition, dummy regression models (eg quantile regression) could give a broader perspective of the effect of the programs.

Minor comments:

- Is Line 107 a paragraph title?

- 72.98+6.49 is understandable if you look at the table but I think it needs to be explained in the text as well. Also, did you want to write 72.98±6.49?

Author Response

Thank you very much for your thorough review. Attached are the corrections we have made based on your review.
